# Explainable artificial intelligence for personalized prognosis in pancreatic cancer: A nationwide study from Taiwan

**Dai-Rong Tsai**[1,2], **Chun-Ju Chiang**[1,2], **Pei-Chun Hsieh**[3], **Chi-Yen Huang**[3], **Wen-Chung Lee**[1,2,4]*

1 Institute of Epidemiology and Preventive Medicine, College of Public Health, National Taiwan University, Taipei, Taiwan, 2 Taiwan Cancer Registry, Taipei, Taiwan, 3 Health Promotion Administration, Ministry of Health and Welfare, Taipei, Taiwan, 4 Institute of Health Data Analytics and Statistics, College of Public Health, National Taiwan University, Taipei, Taiwan

* wenchung@ntu.edu.tw

## Abstract

Pancreatic cancer is highly aggressive with poor outcomes; current artificial intelligence (AI) prognostic models often lack interpretability and underutilize large-scale data. This study develops an explainable AI prognostic model for pancreatic cancer survival using Taiwan's national registry data, aiming to identify key prognostic factors, their interactions, non-linear relationships, and patient-specific survival variability. We analyzed 8,864 pancreatic cancer cases diagnosed between 2013 and 2021 from the Taiwan Cancer Registry. We developed three classes of prognostic models using regression-based, machine learning, and deep learning methods. The models were evaluated using nested cross-validation and time-dependent metrics, with Shapley additive explanations enhancing interpretability. XGBoost outperformed random survival forest and deep learning models in predicting pancreatic cancer survival. Key determinants included surgery, histological type, chemotherapy, tumor stage, and their interactions. Adenocarcinoma was associated with the highest mortality risk, whereas acinar cell and neuroendocrine carcinomas had lower risks (hazard ratios 0.768 and 0.660, respectively, vs adenocarcinoma). Chemotherapy showed the greatest mortality risk reduction in adenocarcinoma, while surgery was most strongly associated with reduced mortality in neuroendocrine tumors and adenocarcinoma, particularly in early-stage disease. The mortality reduction associated with chemotherapy increased in advanced stages and with age, plateauing around 65 years. Mortality risk rose faster with age in neuroendocrine carcinoma. Non-linear relationships emerged for age, smoking duration, and BMI: mortality increased gradually (0.69% per year) before age 65, rapidly afterward (2.41% per year); risk increased with longer smoking duration but plateaued between 10 and 30 years; and BMI exhibited a U-shaped risk, lowest at 26. This study demonstrates the potential of explainable AI for predicting pancreatic cancer survival by identifying key prognostic

**Data availability statement:** Data are available from the National Taiwan Cancer Registry Database published by the Health Promotion Administration, Ministry of Health and Welfare of Taiwan. Due to legal restrictions imposed by the government of Taiwan in relation to the "Personal Information Protection Act", data cannot be made publicly available. Requests for data can be sent as a formal proposal to the Health Promotion Administration, Ministry of Health and Welfare of Taiwan (No.36, Tacheng St., Datong Dist., Taipei, Taiwan, https://www. hpa.gov.tw/EngPages/List.aspx?nodeid=4250). The analysis code and a synthetic dataset with similar structure and statistical properties to the real data were deposited in a public GitHub repository (https://github.com/darentsai/ SurvPancreas).

**Funding:** This work was funded by the Health Promotion Administration, Ministry of Health and Welfare in Taiwan (A1141107), and the National Science and Technology Council in Taiwan (MOST 111-2314-B-002-089-MY3; NSTC 114-2314-B-002-085-MY3). The funders play no role in the study design, data collection and analysis, decision to publish, or preparation of the manuscript.

**Competing interests:** The authors have declared that no competing interests exist.

factors, nonlinear relationships, interactions, and patient-level variability, thereby revealing substantial heterogeneity in prognosis.

## Author summary

Artificial intelligence (AI)-based models have been developed for pancreatic cancer prognosis, but they often lack interpretability, overlook feature interactions and non-linear effects, and are based on limited or single-institution datasets. This study develops an explainable AI prognostic model using Taiwan's nationwide cancer registry, identifying key prognostic factors, non-linear effects, and interactions—especially between treatments and tumor subtypes—while highlighting variability in patient-specific risk. The findings illustrate how explainable AI can be integrated with population-based cancer registry data to improve prognostic assessment and risk stratification in pancreatic cancer, thereby informing the future development of transparent, scalable prognostic tools and methodologies in oncology research and clinical practice.

## Introduction

Pancreatic cancer is an aggressive malignancy with a poor prognosis. In 2022, it accounted for an estimated 467,005 deaths globally, representing 4.8% of all cancer-related deaths and ranking as the 6th leading cause of cancer mortality, with a disproportionately greater burden in high-income countries [1] In Taiwan, pancreatic cancer was the 7th leading cause of cancer death among males and the 5th among females, with age-standardized mortality rates of 6.98 and 5.15 per 100,000, respectively [2]. Approximately two-thirds of patients die within one year of diagnosis, and only about 10% survive beyond five years [3]. Surgical resection followed by adjuvant chemotherapy remains the standard treatment; however, due to nonspecific early symptoms and the absence of effective screening methods, most patients are diagnosed with locally advanced or metastatic tumors that are typically unresectable [4]. Furthermore, survival outcomes can vary considerably even among patients at the same stage and receiving similar treatments. Consequently, developing accurate prognostic models for pancreatic cancer is essential to guide personalized treatment decisions and improve survival predictions.

Several artificial intelligence (AI)-based prognostic models have been developed to predict pancreatic cancer progression and identify critical prognostic factors [5,6]. Commonly used clinical variables include tumor size, metastatic status, lymph node involvement, adjuvant treatment, and molecular biomarkers such as carcinoembryonic antigen (CEA) and carbohydrate antigen (CA) 19–9. Some studies have incorporated genomic features from gene expression profiles to enhance predictive performance [7–9], while others have utilized deep learning algorithms on computed tomography (CT) imaging data to capture dynamic tumor attenuation patterns

and tumor-vascular interactions [10–12]. However, AI-based models are often challenging to interpret. Shapley additive explanations (SHAP) have been proposed as a potential solution to improve interpretability [13–16]. While these studies primarily focus on feature importance and the directionality of their effects, they often overlook heterogeneous effects and interactions among features. Additionally, many studies are based on small hospital datasets with limited geographic coverage, which affects their generalizability and introduces potential selection bias.

This study utilizes Taiwan's nationwide, population-based cancer registry to develop an explainable AI prognostic model for pancreatic cancer survival. We quantify the impact of individual prognostic factors on mortality risk and explore the variability of these effects across patients. Additionally, we identify non-linear relationships and interactions among the prognostic factors.

## Methods

### Ethics statement

This study protocol was approved by the National Taiwan University Research Ethics Committee (NTU-REC No.202405HM031; NTU-REC No.202504HM027) and the Data Release Review Board of the Health Promotion Administration, Ministry of Health and Welfare in Taiwan. All methods were performed in accordance with the relevant guidelines and regulations. The National Taiwan University Research Ethics Committee waived the requirement for informed consent due to the lack of personal information and secondary data in the study. This study does not involve animal subjects.

### Data source

Data on incident pancreatic cancer cases were obtained from the Taiwan Cancer Registry (TCR). TCR is a nationwide, population-based registry recognized globally for its consistently high standards of data quality, completeness, and timeliness [17,18]. We analyzed pancreatic cancer patients diagnosed between January 1, 2013 and December 31, 2021 from the TCR long-form dataset. The dataset provides detailed tumor staging information and covers approximately 50% of all pancreatic cancer cases during that period in Taiwan. Survival outcomes were updated using national death registry data through December 31, 2022. Cases were identified using the International Classification of Diseases for Oncology, Third Edition (ICD-O-3), covering tumors of the pancreatic head (C25.0), body (C25.1), tail (C25.2), and other specified or unspecified subsites (C25.3–C25.9). We excluded patients with carcinoma in situ, those diagnosed with stage 0 or an unspecified stage, those under 20 years of age, those did not receive first course of treatment at the reporting hospital, those without any curative or palliative treatment, and duplicate records. This resulted in a final cohort of 8,864 eligible patients for analysis. The patient inclusion flowchart is shown in S1 Fig.

The primary outcome of this study is overall survival. The potential predictors considered include age at diagnosis, gender, body mass index (BMI), and tumor characteristics such as subsite (head, body, tail, or other parts of the pancreas), histological type, and American Joint Committee on Cancer (AJCC) stage with its components: T-stage (tumor size), N-stage (regional lymph node involvement), M-stage (distant metastasis), and tumor grade (cell differentiation). Histological types were classified according to morphology codes (S1 Table) into adenocarcinoma, neuroendocrine carcinoma, neuroendocrine tumor, solid pseudopapillary neoplasm, acinar cell carcinoma, and other subtypes. Six treatment modalities were assessed: surgery, chemotherapy, radiotherapy, targeted therapy, hormonal therapy, and immunotherapy, each coded as a binary variable to indicate whether the treatment was applied. Additional factors included in the model were behavioral variables such as smoking duration (in years) and alcohol consumption status (never, past, or current), the level of hospitals where patients were diagnosed and treated (medical centers vs. others), and the urbanization level of the patient's residential area, used as a proxy for lifestyle and socioeconomic status [19]. Baseline characteristics of the predictors and outcomes are shown in S2 Table.

## Model development

We developed three classes of prognostic models using regression-based, machine learning, and deep learning techniques. For the regression-based approach, we employed the Cox proportional hazards model in two versions: one including only main effects (Cox) and the other including both main effects and all possible two-way interaction terms (Cox-Interact). To address overfitting and multicollinearity, we applied the group least absolute shrinkage and selection operator (group lasso), a regularization technique that selects groups of related variables as a unit [20]. Coefficients shrunk to zero were eliminated, and the models were subsequently refitted without penalization.

For the machine learning models, we employed both random forest and gradient-boosted trees adapted for censored survival data. A random survival forest builds an ensemble of de-correlated trees using bootstrapped samples and averages their predictions, with each node split determined by maximizing the log-rank statistic [21]. This study used the oblique random survival forest (ORSF), which follows a similar framework but splits data at non-leaf nodes using linear combinations of predictors rather than single variables [22]. For boosting, we applied XGBoost, an optimized implementation of gradient-boosted trees that mitigates overfitting through a regularized learning objective, weight shrinkage, and subsampling of both features and training instances [23]. We specified the Cox partial log-likelihood as the objective function in XGBoost to predict log hazard ratios (HRs) for mortality, and then estimated the baseline hazard using the Breslow method to derive survival probabilities. We also employed three state-of-the-art deep learning survival models: DeepSurv [24], Cox-Time [25], and PC-Hazard [26]. Monotonic network structures were used, defined by three tuning parameters: the number of hidden layers, the ratio of the width of the first hidden layer to that of the input layer, and a common ratio between the widths of consecutive hidden layers, allowing for flexible architectures that can expand, contract, or remain uniform in width across layers.

## Model evaluation

Six performance metrics were adapted to their time-dependent forms to evaluate predictive accuracy at various time points following cancer diagnosis: sensitivity (true positive rate), specificity (true negative rate), precision (positive predictive value), $F_1$ score, area under the ROC curve (AUC, or C-statistic), and scaled Brier score. The $F_1$ score is the harmonic mean of sensitivity and precision. The AUC summarizes sensitivity and specificity across all thresholds and represents the probability that the model assigns a higher risk score to a patient who died compared to one who survived at a given time point. The scaled Brier score reflects the relative reduction in prediction error compared to a null model, indicating the proportion of residual variation explained by the prognostic model. All time-dependent metrics were estimated using inverse probability of censoring weighting to account for information loss due to censoring [27,28]. In addition, we plotted time-dependent calibration curves at multiple time points since cancer diagnosis to assess agreement between predicted risks and observed outcomes.

Before model construction and performance evaluation, we performed multiple imputation by chained equations (MICE) to generate five complete datasets [29]. Missing values in each predictor were imputed by sampling from a conditional distribution estimated using a random forest model based on all other predictors [30,31]. The follow-up time and event indicator were excluded from the imputation process. The MICE algorithm was run for 25 iterations, and convergence was assessed using autocorrelation and the potential scale reduction factor. Autocorrelations approaching zero and scale reduction factors approaching one indicated adequate convergence [29]. The proportion of missing values for each predictor and the corresponding convergence diagnostics are shown in S2 Table and S2 Fig, respectively. For each imputed dataset, we applied nested cross-validation to obtain an unbiased estimate of model performance for selecting the optimal model [32,33]. An outer 5-fold cross-validation loop split the data into five equal subsets, with each (20%) used in turn for evaluation of time-dependent performance metrics, and the remaining (80%) for training. An inner 5-fold cross-validation loop was employed during training for hyperparameter optimization, using a random search strategy [34] over the ranges specified in S3 Table. To reduce variability from data partitioning, we repeated the nested cross-validation process ten

times, resulting in a total of 5(imputed datasets)×5(outer folds)×10(repeats)=250 estimates for each performance metric at each time point. The averages of these estimates were used to summarize the overall trends of each metric over time. The runtimes for training and hyperparameter tuning of all models in each fold are compared in S3 Fig.

**Model interpretation**

Finally, we trained the best-performing model on the full dataset and examined the mechanisms behind its predictions using Shapley additive explanations (SHAP) values [35]. SHAP, a model-agnostic explanation method rooted in game theory, quantifies a feature's marginal contribution by averaging changes in expected model predictions when conditioning on that feature across all possible orderings. SHAP values can also be decomposed into higher-order interaction values distributed across feature pairs, allowing for the explanation of complex interaction effects [36]. Due to the computational complexity of exact SHAP value calculations, KernelSHAP was developed as an efficient and unbiased approximation, fitting a weighted linear regression to feature coalitions with Shapley kernel weights [35,37]. In contrast, TreeSHAP enables exact computation of SHAP values in polynomial time by exploiting the structure of tree-based models [36]. For this study, SHAP values were calculated using TreeSHAP for tree-based models and KernelSHAP for non-tree-based models.

We ranked feature importance and interaction importance by averaging the absolute SHAP values and pairwise SHAP interaction values, respectively, across all training instances. For models with inherent randomness (e.g., XGBoost), features or feature pairs with similar importance scores may have fluctuating rankings across retraining, even when hyperparameters are fixed. To ensure robust inference, we repeatedly trained the model using a set of fine-tuned hyperparameters (listed in S3 Table), and progressively averaged the SHAP interaction values until the importance rankings stabilized. Stability was defined using Kendall's τ correlation coefficient, calculated between successive rankings of the cumulative mean SHAP interaction values. We also derived HRs for mortality for each feature from the SHAP values and quantified their uncertainty using 95% confidence intervals (95% CIs) based on 1,000 bootstrap resamples.

**Results**

Fig 1 shows that performance improves across all models until about 3 years after diagnosis, then plateaus, likely because most deaths occur within the first 3 years (yielding richer signal) while increasing censoring thereafter reduces events for evaluation. XGBoost had the highest sensitivity before year 3 and the highest specificity thereafter. Precision remains consistent across all models, indicating reliable positive predictions. The time-dependent AUC and scaled Brier score curves show widening performance differences over time, with XGBoost consistently achieving the highest values across the full time horizon. The Cox-Interact model does not improve predictive performance over the plain Cox model, except for a slight gain in scaled Brier scores after the third year. Overall, XGBoost emerges as the best-performing model, followed by the three deep learning models and ORSF, while the classical regression approaches (Cox and Cox-Interact) perform the worst. The calibration curves of all competing models at 1, 3, and 5 years after cancer diagnosis are shown in S4 Fig, and the corresponding calibration slopes and intercepts indicate that predicted survival closely matches observed outcomes across models.

We then calculated SHAP values and pairwise SHAP interaction values using the XGBoost model trained on the full dataset, aggregating them separately to rank feature and interaction importance. This process was repeated 15 times and averaged, as additional repetitions did not notably improve the stability of the rankings (S5 Fig). Leveraging XGBoost's built-in tree-based mechanism for handling missing values, the model was trained and SHAP values were computed directly on the original (non-imputed) data. For comparison, we also trained XGBoost models separately on the five imputed datasets and averaged their SHAP values; the resulting importance rankings were highly consistent with those from the original data, with Kendall's τ of 0.95 for feature importance and 0.89 for interaction importance. We therefore adopted the non-imputed approach for subsequent analyses, allowing each feature's effect size to be assessed using SHAP values from patients with complete records for that feature, without reliance on imputed values.

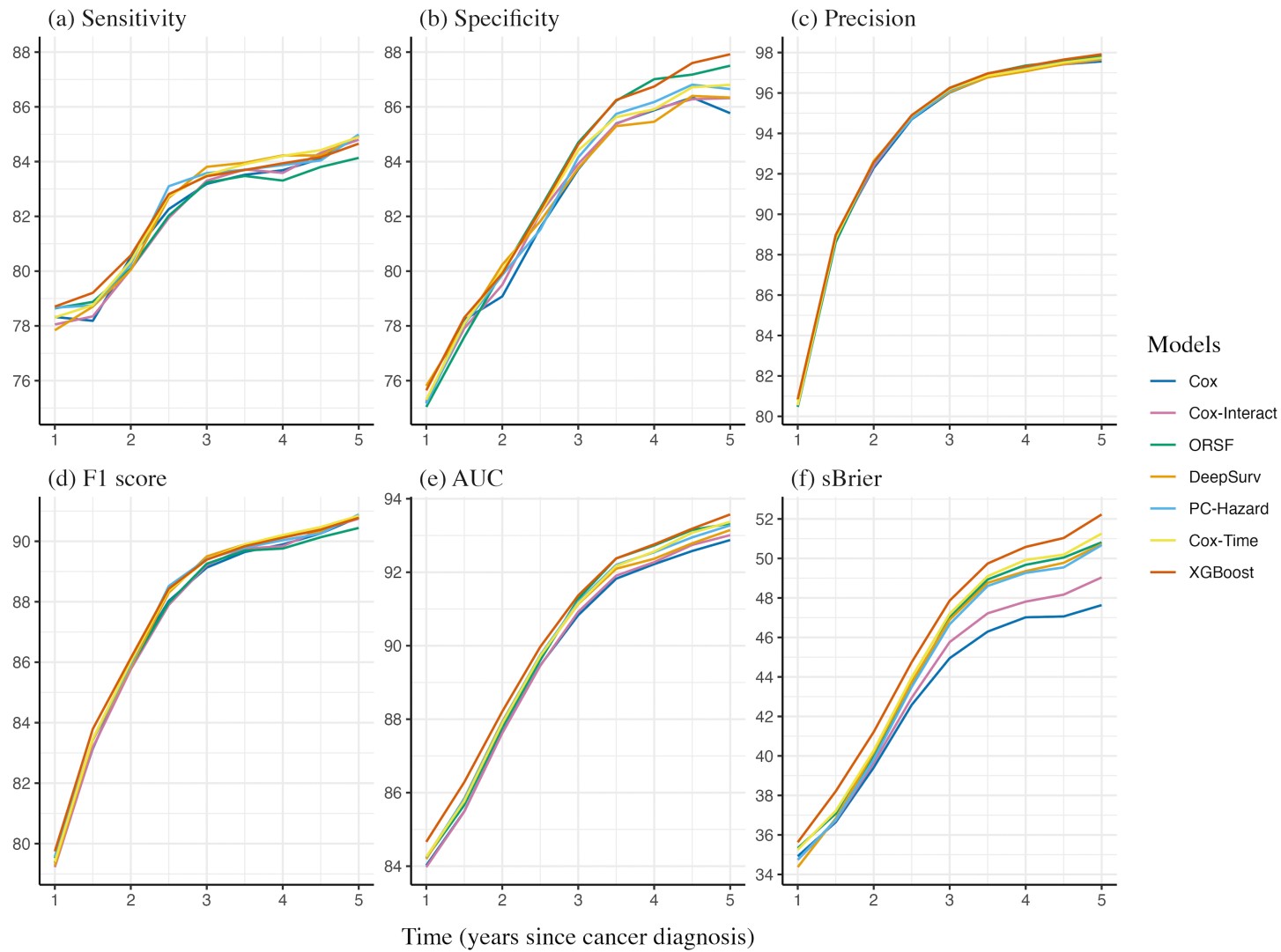

**Fig 1. Time-dependent evaluation metrics for model performance (AUC: area under the receiver operating characteristic curve; sBrier: scaled Brier score; Cox-Interact: interaction-augmented Cox model; ORSF: oblique random survival forest).**

Fig 2 provides an overview of the most important features (left panel) and the feature pairs with the strongest interactions (right panel). A higher interaction importance for a feature pair indicates that the effect of one feature on mortality risk varies more depending on the level of the paired feature. Notably, features such as surgery, histological type, chemotherapy, and AJCC stage are key determinants of survival for pancreatic cancer patients, and they frequently appear among the feature pairs with the highest interaction importance. The SHAP interaction importance is further visualized as a heatmap in S6 Fig, where darker cells represent stronger interactions within feature pairs.

Fig 3 presents a series of SHAP dependence plots for the top-ranked important features, revealing both the global pattern of each feature's main effect on mortality risk and the individual variability in these effects, as shown by the vertical dispersion of SHAP values in the plots. The SHAP values on the y-axis represent the impact of each feature on the model's output, specifically the natural logarithm of the hazard ratio for mortality. Fig 3a,c show that both surgery and chemotherapy—primary treatment modalities for pancreatic cancer—are associated with lower mortality risk in our prognostic

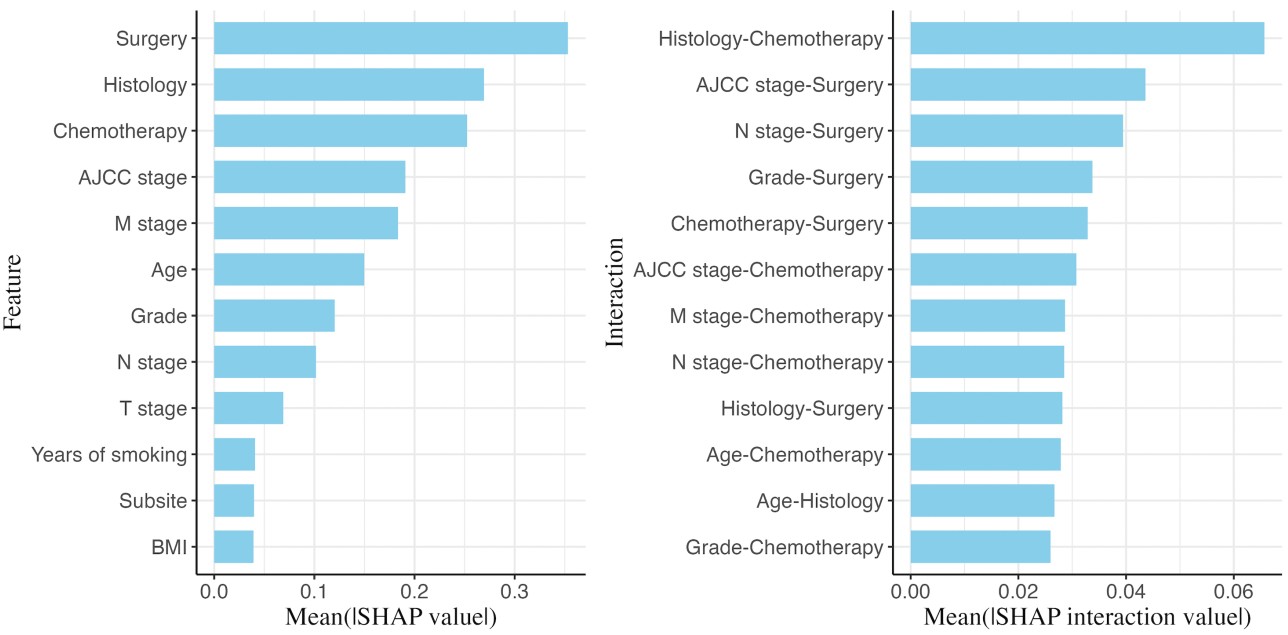

**Fig 2. Global feature importance and interaction importance.** Bar charts of feature and interaction importance measured by averaging the absolute SHAP values and pairwise SHAP interaction values (arranged along the y-axis based on importance; interaction importance multiplied by two).

model. The average SHAP values for patients who underwent surgery and those who did not are −0.464 and 0.295, respectively, indicating that surgical patients have a HR for death of exp(−0.464−0.295)=0.468 (95% CI: 0.467−0.470) compared with non-surgical patients. Similarly, the HR for death is estimated to be exp(−0.145−0.497)=0.526 (95% CI: 0.522−0.531) for patients who received chemotherapy compared with those who did not. Fig 3b shows that different histological types of pancreatic cancer are associated with varied mortality risks: adenocarcinoma is linked to a higher mortality risk, while acinar cell and neuroendocrine carcinomas have risks slightly below the overall average, with HRs of 0.768 (95% CI: 0.730−0.805) and 0.660 (95% CI: 0.643−0.675) relative to adenocarcinoma. In contrast, neuroendocrine tumors and solid pseudopapillary neoplasms are linked to much lower risks, with HRs of 0.188 (95% CI: 0.185−0.191) and 0.165 (95% CI: 0.161−0.169) relative to adenocarcinoma. As expected, tumor-related characteristics such as TNM staging and tumor grade (Fig 3,d,e,g–i) are critical in determining mortality risk, with higher stages and grades correlating with an increased likelihood of death.

Fig 3f,j,l demonstrate XGBoost's ability to capture the nonlinear effects of age, years of smoking, and BMI on mortality risk. The slopes of the SHAP values for age are 0.0068 and 0.0238 before and after age 65, respectively, indicating that the hazard of death increases by [exp(0.0068)−1]×100%=0.69% per year before age 65 (95% CI: 0.66%–0.71%), and by [exp(0.0238)−1]×100%=2.41% per year thereafter (95% CI: 2.39%–2.45%). For smoking duration, mortality risk increases from 0 to 10 years, plateaus between 10 and 30 years, and rises again beyond 30 years. A U-shaped relationship is observed between BMI and mortality risk, with the lowest risk at a BMI of 26; both lower and higher BMI values are associated with increased risk. The high SHAP values at extreme levels of BMI and smoking duration indicate that these features can have a substantial impact for specific individuals, despite their relatively low overall importance. S7 Fig presents the SHAP summary plot, which concisely visualizes the magnitude, distribution, and direction of SHAP values for each feature.

Fig 4 presents SHAP dependence plots for the most strongly interacting feature pairs (interaction values in S8 Fig). The SHAP values of both features in a pair are summed to determine their overall impact. Fig 4a,i highlight the

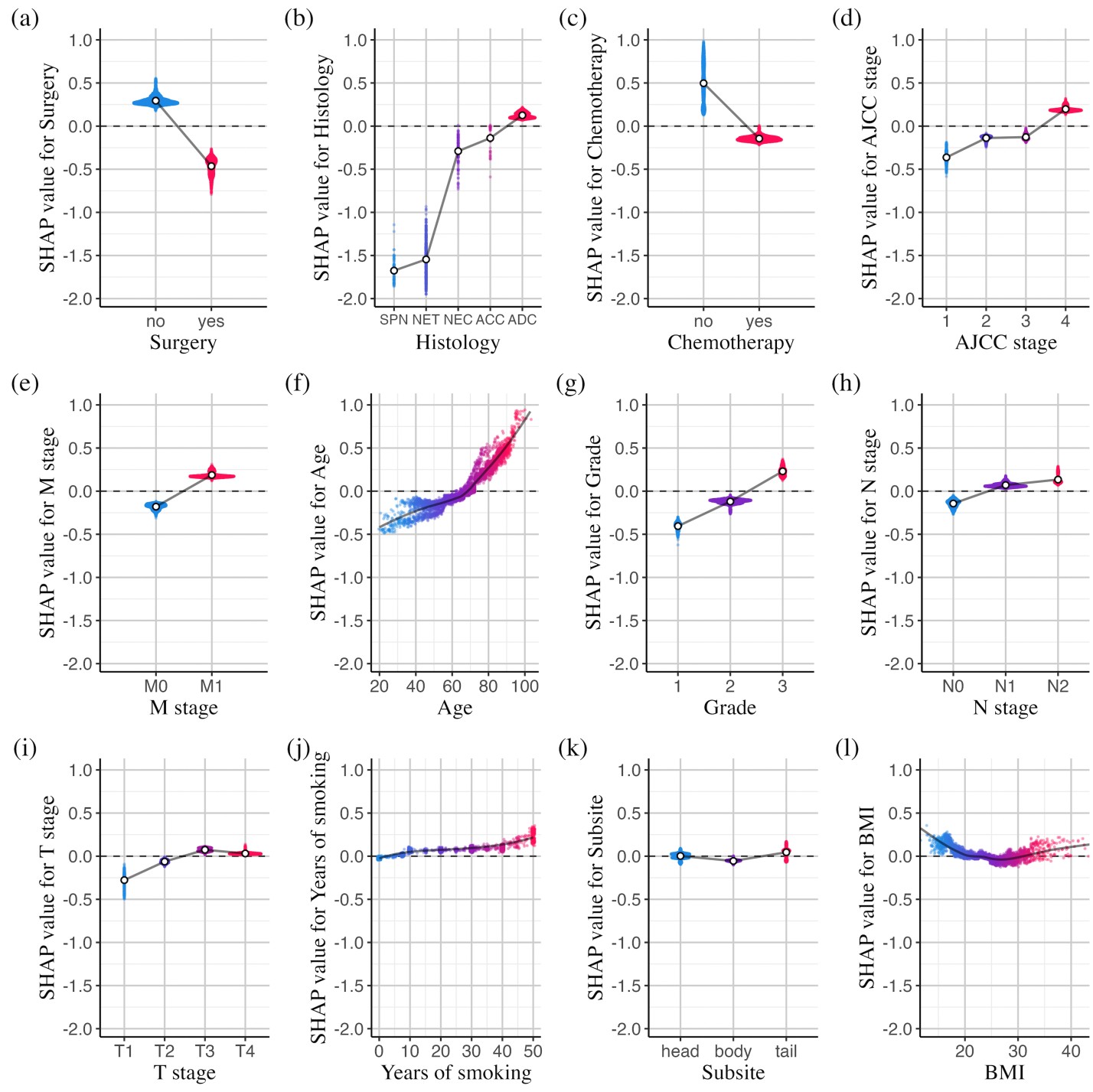

**Fig 3. SHAP dependence plots for the 12 most important features (dots for categorical variables: positions of the average SHAP values at each level; trend lines for continuous variables: fitted using locally estimated scatterplot smoothing; ADC: adenocarcinoma; NEC: neuroendocrine carcinoma; NET: neuroendocrine tumor; SPN: solid pseudopapillary neoplasm; ACC, acinar cell carcinoma).**

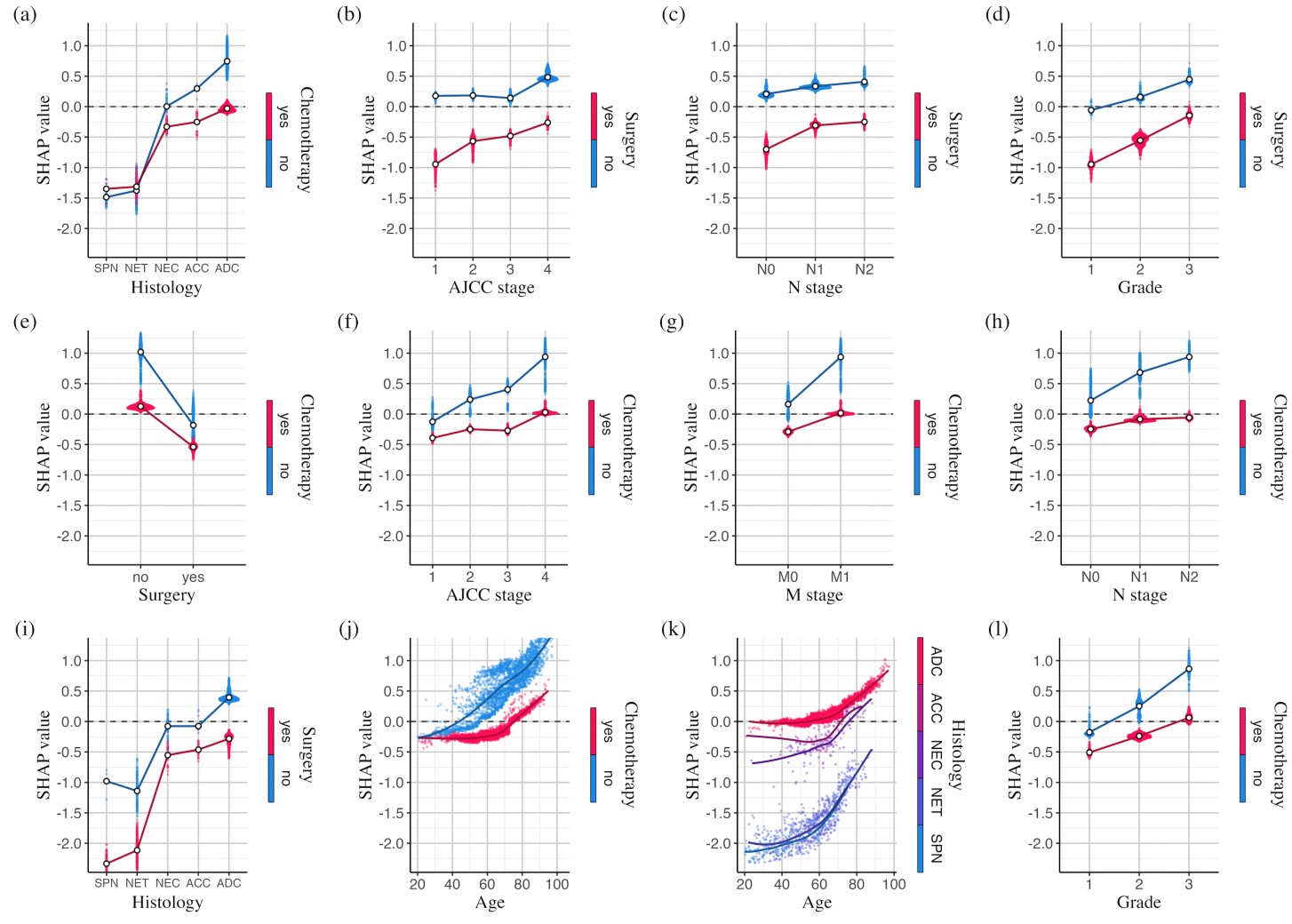

**Fig 4. SHAP dependence plots for the 12 most strongly interacting feature pairs (dots for categorical variables: positions of the average SHAP values at each level; trend lines for continuous variables: fitted using locally estimated scatterplot smoothing; ADC: adenocarcinoma; NEC: neuroendocrine carcinoma; NET: neuroendocrine tumor; SPN: solid pseudopapillary neoplasm; ACC: acinar cell carcinoma).**

differential impacts of chemotherapy and surgery across histological types. Chemotherapy is associated with the greatest reduction in mortality risk for adenocarcinoma (HR = 0.460; 95% CI: 0.455–0.464) and has only a limited effect in neuroendocrine tumors, whereas surgery is associated with particularly low mortality risk in neuroendocrine tumors (HR = 0.378; 95% CI: 0.363–0.393) and adenocarcinoma (HR = 0.508; 95% CI: 0.506–0.509). Nearly all patients with solid pseudopapillary neoplasms underwent surgery without chemotherapy, making it difficult to assess separate treatment effects for this subtype. Tumor characteristics further interact with treatment modalities, with surgery associated with the most substantial reduction in mortality in milder cases (Fig 4b–d), whereas chemotherapy shows the strongest association with reduced mortality in advanced stages (Fig 4f–h, l). Additionally, Fig 4j shows a clear change in the association between chemotherapy and mortality over the life course. Among patients who did not receive chemotherapy, mortality risk increases with age by 2.33% per year (95% CI: 2.28%–2.38%), whereas in those receiving chemotherapy, this rise is attenuated to 0.24% per year (95% CI: 0.23%–0.26%) until age 65, after which it accelerates

to 2.60% per year (95% CI: 2.57%–2.63%). Consistent with this pattern, the mortality reduction associated with chemotherapy strengthens with age and plateaus around 65 years (HR = 0.489; 95% CI: 0.481–0.497). Fig 4k shows that mortality risk rises faster with age for neuroendocrine carcinoma, neuroendocrine tumors, and solid pseudopapillary neoplasms than for adenocarcinoma or acinar cell carcinoma, especially in older patients. Additional SHAP dependence plots for feature pairs ranked 13–20 are provided in S9 Fig.

## Discussion

Theoretically, a log-linear Cox model allows for the explicit inclusion of nonlinear, time-dependent, and interaction terms. However, specifying these terms requires extensive exploratory analysis and statistical testing, increasing the risk of model mismatch if the assumed structure does not accurately capture the true relationships in the data [36]. This study incorporates both main effects and all possible two-way interaction terms, applying group lasso regularization for feature selection to enhance generalizability. Despite this adjustment, the Cox-Interact model does not show improved predictive performance compared to the plain Cox model.

In contrast, machine learning and deep learning methods can automatically capture complex nonlinear relationships and interactions in the data. When predicting pancreatic cancer survival, all AI-based prognostic models outperformed the Cox models across multiple performance metrics, with XGBoost emerging as the top performer and showing a clear advantage over the three deep learning models. Extensive benchmarks have demonstrated that tree-based models consistently outperform neural networks on tabular datasets, where features are individually meaningful and lack strong spatial or temporal dependencies [38]. Although our findings in real-world pancreatic cancer survival data are consistent with this pattern, it may partly reflect the relatively simple neural network architectures used in our study. In addition, when compared with recent pancreatic cancer prognostic models [7–16], our XGBoost model achieved superior AUCs across the 1- to 5-year horizons (S4 Table).

Using an AI prognostic model enhanced with Shapley explanations, this study highlights notable nonlinear effects of age, smoking, and BMI on pancreatic cancer survival. The increase in mortality risk accelerates after age 65, which may relate to declining physiological reserve, accumulating comorbidities, and age-dependent tumor biology. Cigarette smoking is associated with reduced survival in pancreatic cancer patients, while smoking cessation may improve prognosis [39]. In our analyses, mortality risk increases with longer smoking duration but appears to plateau between 10 and 30 years, which may reflect survivor bias and competing risks. Additionally, both low and high BMI levels are associated with increased pancreatic cancer mortality [40,41]. Low BMI may reflect nutrient depletion or cachexia, whereas high BMI is indicative of metabolic dysfunction and chronic inflammation. In contrast, although previous studies have reported an association between alcohol consumption and decreased pancreatic cancer survival [42], this study finds no significant difference in mortality risk between drinkers and non-drinkers.

This study reveals important interactions between treatment modalities and histological subtypes of pancreatic cancer. Chemotherapy was associated with the greatest reduction in mortality risk for adenocarcinoma but had only a limited effect in neuroendocrine tumors. In contrast, surgery was associated with particularly low mortality risk in both neuroendocrine tumors and adenocarcinoma. Furthermore, the results highlight important interactions between tumor characteristics and treatment modalities: surgery is linked to the largest mortality reduction in early-stage disease, whereas chemotherapy shows a stronger association with reduced mortality in advanced stages. In addition, the mortality reduction associated with chemotherapy increases with age and appears to peak around 65 years. These interactions illustrate substantial heterogeneity in mortality risk across tumor stage, histological type, and patient age. Finally, the mortality risk was found to rise more rapidly with age in patients with neuroendocrine carcinoma, neuroendocrine tumors, and solid pseudopapillary neoplasms compared to adenocarcinoma or acinar cell carcinoma, indicating that these subtypes may have different age-related mortality risks.

Our explainable AI model could support clinical decision-making in several ways. First, it may assist risk stratification by identifying patients at elevated risk who could benefit from closer monitoring, earlier integration of palliative care, or

consideration of more intensive treatment options. The treatment-related interactions revealed by the model also help characterize how prognosis varies across clinical subgroups. For example, our findings suggest that older patients with advanced-stage adenocarcinoma constitute a particularly high-risk group in whom the mortality reduction associated with chemotherapy appears most pronounced, a pattern that may warrant further investigation in dedicated clinical or observational studies.

Second, patient-level SHAP values enable transparent, individualized risk communication: clinicians can show which factors contribute most to a specific patient's predicted risk and how changes in key variables (e.g., smoking duration, BMI) might influence prognosis. S10 Fig illustrates clinical vignettes for three example patients with distinct characteristics, displaying their input features, corresponding SHAP values, predicted survival curves, and possible clinical interpretation. Finally, several implementation issues would need to be addressed before clinical deployment, including integration with electronic health record systems, development of user-friendly interfaces, clinician training on model interpretation, and ongoing surveillance for performance drift over time. Together, these elements outline a potential pathway for translating our model into a practical clinical decision-support tool, while recognizing that its role should complement, rather than replace, clinical judgment and guideline-based care.

One key limitation is the observational design, as treatment–outcome associations may be influenced by residual confounding, confounding by indication, and treatment selection bias (e.g., unmeasured differences in performance status, frailty, or disease progression between treated and untreated patients). Accordingly, the observed effects of surgery and chemotherapy represent prognostic associations rather than causal estimates and should be interpreted with caution; these associations should be validated in randomized trials or rigorously controlled observational studies before being used to inform treatment allocation. In addition, because the model is prognostic rather than prescriptive, it does not generate treatment recommendations, and we did not assess concordance between model outputs and clinical treatment decisions.

Second, treatments were coded as baseline "ever/never" variables, which may introduce immortal-time bias because patients must survive long enough after diagnosis to receive surgery or chemotherapy; this can overstate apparent associations by attributing pre-treatment survival time to the treated group. Future analyses should model treatments as time-dependent covariates or use landmark analyses to better align exposure definition with clinical timing. Third, we did not include diagnosis year (2013–2021) as a covariate and therefore did not assess whether treatment–outcome associations varied across calendar time; future work could incorporate diagnosis year and evaluate treatment-by-year heterogeneity, including via SHAP interaction patterns, to examine temporal variation in these associations.

Methodologically, imputing before cross-validation may leak information via validation-fold covariate distributions and inflate performance. Although follow-up time and event status were excluded to prevent outcome-driven leakage, future analyses should nest imputation within training folds; for deployment, imputation models should be trained on development data and applied to new patients. In addition, because we highlighted top features and interactions from a larger set, the analysis is subject to multiplicity and potential selection effects; thus, these findings should be viewed as exploratory and hypothesis-generating rather than confirmatory.

Moreover, key prognostic factors, including diabetes history, glycated hemoglobin (HbA1c), and tumor markers such as pre-treatment CEA and CA19–9, have only been recorded in the Taiwan Cancer Registry since 2021 and were unavailable for the main analysis. In a sensitivity analysis of the 2021–2023 cohort (≈90% case coverage; 7,336 patients) using the same cross-validation workflow, adding these biomarkers modestly improved the 1-year AUC from 0.855 to 0.865 and the 3-year AUC from 0.890 to 0.895 (S11 Fig), suggesting scope for further gains; future studies incorporating these biomarkers alongside genomic features (e.g., KRAS, TP53 mutations) and imaging data may further improve prediction and reveal additional interaction patterns. Finally, external validation in independent cohorts remains essential: temporal validation in 2022–2023 yielded 1-year and 2-year AUCs of 0.858 and 0.878, respectively, but geographic and cross-ethnic validation is still needed to establish generalizability across populations and healthcare settings.

## Conclusion

This study developed an AI-based prognostic model for pancreatic cancer using Taiwan's nationwide cancer registry data. By integrating explainable artificial intelligence, we identified key prognostic factors, their nonlinear relationships and interactions, and the variability of these associations across patients, thereby highlighting substantial heterogeneity in pancreatic cancer prognosis.

## Supporting information

**S1 Table. ICD-O-3 morphology codes for pancreatic cancer histological types.**
(PDF)

**S2 Table. Baseline characteristics and outcomes of pancreatic cancer patients.**
(PDF)

**S3 Table. Hyperparameters and tuning ranges.**
(PDF)

**S4 Table. Comparison with recent pancreatic cancer prognostic models.**
(PDF)

**S1 Fig. Study flowchart.**
(PDF)

**S2 Fig. MICE convergence diagnostics.**
(PDF)

**S3 Fig. Distribution of runtimes for model training and hyperparameter tuning.**
(PDF)

**S4 Fig. Time-dependent calibration curves of the competing models.**
(PDF)

**S5 Fig. Stability of the SHAP interaction importance rankings.**
(PDF)

**S6 Fig. Heatmap of SHAP interaction importance matrix.**
(PDF)

**S7 Fig. SHAP summary plot for the 12 most important features.**
(PDF)

**S8 Fig. SHAP interaction values for the 12 most strongly interacting feature pairs.**
(PDF)

**S9 Fig. SHAP dependence plots for feature pairs ranked 13–20.**
(PDF)

**S10 Fig. Clinical vignettes of three example patients.**
(PDF)

**S11 Fig. Time-dependent AUCs from sensitivity analysis.**
(PDF)

## Author contributions

**Conceptualization:** Wen-Chung Lee.

**Data curation:** Chun-Ju Chiang, Pei-Chun Hsieh, Chi-Yen Huang.

**Formal analysis:** Dai-Rong Tsai.

**Funding acquisition:** Wen-Chung Lee.

**Investigation:** Wen-Chung Lee.

**Methodology:** Dai-Rong Tsai, Wen-Chung Lee.

**Project administration:** Chun-Ju Chiang.

**Resources:** Chun-Ju Chiang, Pei-Chun Hsieh, Chi-Yen Huang.

**Software:** Dai-Rong Tsai.

**Supervision:** Wen-Chung Lee.

**Validation:** Wen-Chung Lee.

**Visualization:** Dai-Rong Tsai, Wen-Chung Lee.

**Writing – original draft:** Dai-Rong Tsai.

**Writing – review & editing:** Wen-Chung Lee.

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
