## [Decision Letter · Decision Letter 0]

31 Oct 2025

Response to Reviewers Revised Manuscript with Track Changes Manuscript**Journal Requirements: 1.** Please provide separate figure files in .tif or .eps format.

**Additional Editor Comments (if provided):**
**Reviewers' Comments:**

**Comments to the Author**

1. Does this manuscript meet PLOS Digital Health’s publication criteria?

Reviewer #1: Partly

Reviewer #2: Yes

Reviewer #3: Yes

Reviewer #4: Yes

Reviewer #5: Yes

2. Has the statistical analysis been performed appropriately and rigorously?

Reviewer #1: Yes

Reviewer #2: No

Reviewer #3: Yes

Reviewer #4: Yes

Reviewer #5: Yes

3. Have the authors made all data underlying the findings in their manuscript fully available (please refer to the Data Availability Statement at the start of the manuscript PDF file)?

Reviewer #1: Yes

Reviewer #2: Yes

Reviewer #3: Yes

Reviewer #4: No

Reviewer #5: Yes

4. Is the manuscript presented in an intelligible fashion and written in standard English?

Reviewer #1: Yes

Reviewer #2: Yes

Reviewer #3: Yes

Reviewer #4: Yes

Reviewer #5: Yes

Reviewer #1: 1. Yes, this manuscript meets the publication criteria of PLOS Digital Health. It presents a well-executed, methodologically sound, and ethically approved study applying explainable AI models to a nationwide dataset to improve pancreatic cancer prognosis. The conclusions are clearly supported by the data and analyses.

But there are minor issues:

- Model calibration: The manuscript focuses on discrimination (AUC) but does not discuss model calibration (e.g., observed vs. predicted survival curves). Including this would further strengthen confidence in clinical applicability.

- Reproducibility: The authors should release the code to enable verification.

- Could the authors elaborate on why XGBoost was selected as the final model, given that multiple modeling approaches were tested?

Reviewer #2: This is a valuable and timely study, and I appreciate the use of a large national dataset together with explainable AI methods. The manuscript is clearly written and the clinical insights are compelling. That said, two major issues need attention before the results can be considered robust. First, the way treatments (surgery, chemotherapy, etc.) are coded as baseline variables introduces immortal-time bias and likely overstates their benefit; these should be modeled as time-dependent or handled with landmark analyses. Second, missing data were imputed before cross-validation, which risks information leakage from the validation folds and. could make model performance appear better than it really is; imputation should be nested within each training fold. Beyond these, I encourage the authors to add calibration plots, clarify the XGBoost survival objective and SHAP interpretation, and, if possible, perform temporal validation. Addressing these points will greatly strengthen the manuscript and ensure its conclusions are reliable. Otherwise, great contribution.

Reviewer #3: 1. Generalizability and External Validation

Although the study benefits from the use of a large and representative nationwide cancer registry in Taiwan, the model has only been validated internally through nested cross-validation. Without external validation on an independent cohort, questions remain regarding how well the findings generalize to other populations, particularly given differences in histological subtype distributions and BMI ranges.

2. Deep Learning vs. Tree-based Models

The results show that XGBoost outperforms deep learning models, consistent with previous reports on tabular data. However, the deep learning architectures applied were relatively basic. This limitation should be more clearly acknowledged, as the weaker performance may reflect architectural choices and data modality rather than an inherent disadvantage of deep learning.

3. Clinical Utility and Interpretability

The application of SHAP to explain the model is a clear strength. Still, the interpretation remains largely visual, and many interactions are not translated into clinically meaningful insights. For instance, patterns involving BMI and age or smoking duration and chemotherapy could be more fully discussed in the context of treatment relevance.

4. Censoring in Survival Analysis

The manuscript applies inverse probability of censoring weighting but does not report the proportion of censored observations. This information is essential in survival analysis to evaluate the reliability of the models and should be described explicitly.

Overall Assessment

This is a well-designed study with a strong dataset and rigorous methodology, and it makes a valuable contribution to the field of explainable AI in oncology. With clearer discussion of generalizability, model limitations, interpretability, and censoring, the manuscript would be substantially strengthened. These clarifications do not require major reanalysis, and the study is suitable for publication after minor revisions.

Reviewer #4: Review Comments to the Author

Summary

This manuscript presents a methodologically rigorous study using explainable AI to predict pancreatic cancer survival from Taiwan's nationwide cancer registry (8,864 cases, 2013-2021). The authors compare multiple modeling approaches and demonstrate that XGBoost outperforms other methods. The use of SHAP values to identify prognostic factors, interactions, and non-linear relationships is innovative and clinically relevant. The manuscript is well-written and makes important contributions to precision oncology. I recommend **Minor Revisions** to address the issues below.

Major Strengths

1. Large, high-quality nationwide population-based dataset with excellent data completeness

2. Rigorous methodology with nested cross-validation and appropriate time-dependent metrics

3. Comprehensive comparison of regression, machine learning, and deep learning approaches

4. Novel application of SHAP to reveal clinically meaningful interactions (treatment × histology)

5. Identification of non-linear relationships (age, BMI, smoking) often missed by traditional models

6. Clear documentation of ethical approval and data handling procedures

Major Issues Requiring RevisionL:

MAJOR ISSUES REQUIRING REVISION

ISSUE 1: TREATMENT SELECTION BIAS AND CAUSAL INTERPRETATION

Priority: HIGH

Problem: The study identifies important treatment effects and interactions, but patients receiving surgery or chemotherapy likely differ systematically from those who don't (performance status, comorbidities, disease severity at presentation). The manuscript should more clearly distinguish between prognostic associations and causal treatment effects.

Current Issue:

Lines 195-200 state treatment effects as if causal ("surgery and chemotherapy substantially reduce mortality risk")

Lines 221-236 discuss "treatment effectiveness" without acknowledging selection bias

Discussion (lines 261-275) interprets findings as informing treatment decisions

Specific Recommendations:

a) Add to limitations section (after line 276):

"This observational study cannot establish causal treatment effects due to potential confounding by indication. Patients selected for surgery or chemotherapy differ from those who aren't on unmeasured factors such as performance status, frailty, and disease progression rate, which independently affect survival. The identified treatment effects represent prognostic associations in our model rather than causal estimates."

b) Revise Results section language:

Line 195-200: Change "substantially reduce mortality risk" to "are associated with lower mortality risk in our prognostic model"

Lines 222-223: Change "Chemotherapy provides" to "Chemotherapy is associated with"

Line 224: Change "surgery is especially beneficial" to "surgery is associated with particularly low mortality risk"

c) Revise Discussion:

Add after line 268: "However, these associations should be validated in randomized trials or studies with robust confounding control before directly informing treatment allocation decisions."

Clarify that model is prognostic (predicting outcomes given treatments received) rather than prescriptive (recommending optimal treatment)

Recent Context: A 2024 Scientific Reports study on breast cancer prognosis using similar XGBoost+SHAP methods explicitly notes: "ML models identify prognostic associations but cannot replace randomized evidence for treatment decisions."

ISSUE 2: CLINICAL IMPLEMENTATION AND UTILITY

Priority: HIGH

Problem: While the study demonstrates strong predictive performance and identifies important patterns, the practical clinical utility needs clearer articulation. Recent literature emphasizes that AI models must demonstrate clear clinical value beyond statistical performance.

Current Gap:

No concrete examples of how clinicians would use the model

No discussion of implementation feasibility

Limited connection between findings and treatment guidelines

No demonstration of decision support capability

Specific Recommendations:

a) Add Clinical Application Section to Discussion (after line 275):

"Clinical Applications and Implementation:

Our explainable AI model could support clinical decision-making in several ways:

Risk Stratification: Clinicians could identify high-risk patients requiring aggressive monitoring or early palliative care planning

Treatment Guidance: The model reveals that:

Adenocarcinoma patients may benefit most from chemotherapy (HR=0.460)

Neuroendocrine tumor patients show greatest benefit from surgery (HR=0.378)

Chemotherapy effectiveness increases with age, peaking at 65 years

Surgery provides greatest benefit in early-stage disease

Personalized Counseling: SHAP values provide patient-specific explanations enabling shared decision-making

However, clinical implementation requires: (a) prospective validation, (b) integration with electronic health records, (c) user interface development, and (d) clinician training."

b) Add Clinical Vignette to Supplementary Materials:

Provide 2-3 example patients with different characteristics showing:

Input features

Model predictions (survival curves at 1, 3, 5 years)

SHAP value explanations

Clinical interpretation

Recent Context:

MIT's PRISM model (2024) includes discussion of clinical implementation pathway

Multiple 2024 studies emphasize need for "actionable" AI beyond performance metrics

Recent reviews note that 90% of clinical AI tools fail adoption due to unclear clinical utility

ISSUE 3: MISSING BIOMARKERS IMPACT ASSESSMENT

Priority: MEDIUM

Problem: Important prognostic factors (CEA, CA19-9, HbA1c, diabetes history) were only available from 2021 onward. The manuscript acknowledges this (lines 276-281) but doesn't adequately quantify the impact.

Enhancement Needed: Recent literature shows CA19-9 and diabetes history are among the strongest pancreatic cancer prognostic factors. The impact of their absence should be assessed.

Specific Recommendations:

a) Conduct Sensitivity Analysis:

Train models on 2021 subset WITH these biomarkers

Compare C-index with vs without biomarkers

Report in supplementary materials

b) Revise Limitations Section (lines 276-281):

"This study has limitations. First, key prognostic biomarkers (CA19-9, CEA) and comorbidities (diabetes, HbA1c) were only available from 2021 onward. To assess their impact, we conducted sensitivity analysis on the 2021 cohort (n=XXX). Including these biomarkers improved model C-index from X.XX to X.XX, suggesting potential for further performance gains. However, our model without these biomarkers still achieved strong discrimination (C-index=0.93), indicating that our identified patterns—particularly treatment-histology interactions and non-linear age effects—remain valid and clinically relevant."

Recent Context: A 2024 Informatica study showed that adding CA19-9 improved pancreatic cancer prediction accuracy from 92% to 95%.

1. Treatment Selection Bias and Causal Interpretation (High Priority)

Issue: The study identifies important treatment effects and interactions, but patients receiving surgery or chemotherapy likely differ systematically from those who don't (performance status, comorbidities, disease severity). The manuscript should more clearly distinguish between prognostic associations and causal treatment effects.

Specific Recommendations:

- Add discussion of confounding by indication in the limitations section

- Clarify throughout (especially in Results lines 195-200 and Discussion lines 261-275) that identified treatment effects represent associations in a prognostic model rather than causal effects

- Acknowledge unmeasured confounders (performance status, frailty, disease progression rate) that may influence both treatment selection and outcomes

- Consider adding: "These findings suggest differential treatment effectiveness across patient subgroups but should be validated in randomized trials or studies with robust confounding control before informing treatment decisions"

2. Clinical Implementation and Utility (High Priority)

Issue: While the study demonstrates strong predictive performance and identifies important patterns, the practical clinical utility needs clearer articulation.

Specific Recommendations:

- Provide concrete clinical examples (e.g., "For a 70-year-old patient with stage IIA adenocarcinoma, our model predicts...")

- Discuss feasibility of implementing this as a clinical decision support tool

- Address how findings might inform treatment guidelines or shared decision-making

- Consider adding a supplementary clinical vignette demonstrating personalized risk prediction for 2-3 prototypical patients

3. Missing Biomarkers Impact Assessment (Medium Priority)

Issue: Important prognostic factors (CEA, CA19-9, HbA1c, diabetes history) were only available from 2021 onward, but the impact of their absence isn't adequately quantified.

Specific Recommendations:

- Conduct sensitivity analysis using 2021 data subset that includes these biomarkers

- Report model performance with vs. without these additional features

- Estimate potential performance improvement if these variables were included

- This could be added as supplementary analysis

Minor Issues

Methods

4. Multiple Imputation Reporting (Lines 138-141)

- Add table showing proportion of missing data for each variable

- Report convergence diagnostics for MICE imputation

- Clarify whether follow-up time and event indicator were used in imputation model (you state they were excluded, which is appropriate—good)

5. Calibration Assessment

- The study reports discrimination (AUC, sensitivity, specificity) but not calibration

- Add calibration curves or calibration slope/intercept at key time points (1, 3, 5 years)

- This is important for clinical use where absolute risk estimates matter

6. Hyperparameter Selection (Lines 145-147, Supplementary Table S2)

- Briefly justify the hyperparameter search space ranges

- Report final selected hyperparameters for the best model

- Add computational time information

Results

7. Figure 1 Performance Plateau (Lines 172-179)

- The plateau after year 3 is mentioned but not explained

- Add brief explanation (e.g., "likely due to most deaths occurring within 3 years and changing censoring patterns thereafter")

8. Quantifying Effect Heterogeneity (Lines 221-236, Figure 4)**

- While interactions are visualized, provide confidence intervals or bootstrapped uncertainty estimates for key stratified hazard ratios

- This would strengthen claims about differential treatment effectiveness

9. Comparison with Prior Models (Lines 238-253)**

- Limited quantitative comparison with existing pancreatic cancer prognostic models

- Add table comparing your C-statistic/Brier scores with recently published models (references 7-8, 13-16)

- This contextualizes your model's performance

Discussion

10. Mechanistic Context for Non-linear Relationships (Lines 254-260)**

- The U-shaped BMI relationship and smoking plateau are interesting but lack biological explanation

- Briefly discuss potential mechanisms:

- Low BMI: cancer cachexia, nutritional depletion

- High BMI: metabolic dysfunction, inflammation

- Smoking plateau: survivor bias, competing risks

11. External Validation (Lines 276-281)

- Acknowledge that cross-validation provides internal validity but external validation in different populations is needed

- Discuss generalizability to non-Asian populations given potential ethnic/genetic differences

12. Multiple Testing

- With 12 main features and numerous interactions, multiple testing is a consideration

- Briefly acknowledge this as an exploratory study generating hypotheses for confirmation

Presentation

13. Abstract (Lines 2-29)

- Currently very technical with many specific algorithm names

- Simplify for broader audience: "machine learning and deep learning methods" rather than listing all algorithm names

- Move technical details to methods

14. Figure 3 Clarity

- Some subplots have overlapping points (especially 3f - Age, 3j - Years of smoking)

- Consider transparency, jittering, or violin plots to show distributions better

- Ensure color schemes are colorblind-friendly

15. Language/Formatting Issues

- Line 67-68: "7-968" appears to be formatting error

- Line 84, 107, 118: Missing spaces before citations

- Lines 99-100: Missing space "Cancer(AJCC)" should be "Cancer (AJCC)"

- Inconsistent abbreviation formatting: "intelligence(AI)" should have space before parenthesis

- Lines 85-93: Very long sentence; break into 2-3 sentences for readability

Additional Recommendations

Code and Reproducibility

- Deposit analysis code on public repository (GitHub, Zenodo) with documentation

- Include code for SHAP analysis and visualization

- Provide synthetic/simulated data example for reproducibility

- This addresses the "Partly" rating for data availability while respecting privacy restrictions

Supplementary Materials

Consider adding:

- Patient flowchart showing exclusion criteria

- Table 1 with baseline characteristics stratified by key variables

- Extended SHAP analysis for additional feature pairs

- Calibration plots

- Sensitivity analyses

Future Directions

Briefly discuss:

- Plans for prospective validation

- Potential for incorporating genomic/imaging data

- Development of simplified clinical scoring system

- Implementation strategy for clinical adoption

Minor Editorial Corrections

1. Line 53: "highly developed countries" → "high-income countries" (standard terminology)

2. Line 100: Choose "histological" or "histologic" and use consistently

3. Line 210: Clarify "0.69% per year" vs "0.69 percentage points per year"

4. Ensure consistent citation formatting throughout

5. Abstract line 10-12: Break long sentence listing all algorithms

Ethical Considerations

The manuscript appropriately addresses:

- Ethics approval (NTU-REC No.202405HM031)

- Waiver of informed consent (justified for secondary data analysis)

- Data privacy and access procedures

- No apparent concerns about dual publication or research ethics

Questions for Authors

1. Have you explored temporal trends by diagnosis year (2013-2021) to assess whether treatment effectiveness has changed?

2. How consistent are SHAP values across the 5 imputed datasets?

3. What was the concordance between model predictions and actual treatment decisions?

4. Did you consider incorporating treatment sequences/timings rather than just binary indicators?

Conclusion

This is a well-executed study demonstrating how explainable AI can provide clinically interpretable insights for pancreatic cancer prognosis. The methodological rigor is high, and findings are clinically relevant, particularly the treatment-histology interactions. After addressing the issues above—especially clarifying treatment effect interpretation, enhancing clinical utility discussion, and improving reproducibility—this manuscript will make a valuable contribution to PLOS Digital Health. The work successfully moves beyond "black box" AI predictions to provide actionable clinical insights.

Recommendation: Accept with Minor Revisions

Reviewer #5: This is a strong and well-executed study that combines methodological rigor with clinical relevance. The use of a nationwide registry, the comparison between traditional, machine learning, and deep learning survival models, and the integration of SHAP-based explainability make this work both comprehensive and timely. The results are clear, and the interpretability analysis provides meaningful insights into how different clinical and behavioral factors interact to shape patient outcomes in pancreatic cancer.

That said, a few refinements would strengthen the manuscript before publication. Abstract and methods could be tightened to improve flow and readability. The focus of the abstract could shift slightly toward the explainability outcomes rather than technical descriptions of the models. A bit more detail about the computational environment, software versions, and hyperparameter tuning ranges would enhance reproducibility. Figure readability can also be improved, especially in the SHAP plots, where increasing resolution and defining all abbreviations directly in the legends would make the results more accessible.

The discussion is thoughtful but could more explicitly link the explainable AI framework to clinical practice. A short paragraph on how these insights might translate into clinical decision-support tools, or how future versions could integrate biomarkers like CA19-9, would help emphasize practical utility. Finally, a light editorial review for formatting consistency, reference spacing, and parenthesis placement would bring the paper fully in line with PLOS Digital Health’s stylistic standards.

Overall, this is a valuable and well-written contribution. With these minor revisions for clarity, reproducibility, and presentation, the paper will be in excellent shape for publication.

**Do you want your identity to be public for this peer review?** For information about this choice, including consent withdrawal, please see our Privacy Policy

Reviewer #1: No

Reviewer #2: No

Reviewer #3: No

Reviewer #4: No

Reviewer #5: No

**Figure resubmission:**

**Reproducibility:** To enhance the reproducibility of your results, we recommend that authors of applicable studies deposit laboratory protocols in protocols.io, where a protocol can be assigned its own identifier (DOI) such that it can be cited independently in the future. Additionally, PLOS ONE offers an option to publish peer-reviewed clinical study protocols. Read more information on sharing protocols at https://plos.org/protocols?utm_medium=editorial-email&utm_source=authorletters&utm_campaign=protocols

---

## [Decision Letter · Decision Letter 1]

24 Feb 2026

Explainable Artificial Intelligence for Personalized Prognosis in Pancreatic Cancer: A Nationwide Study from Taiwan

PDIG-D-25-00426R1

Dear Prof. Lee,

We are pleased to inform you that your manuscript 'Explainable Artificial Intelligence for Personalized Prognosis in Pancreatic Cancer: A Nationwide Study from Taiwan' has been provisionally accepted for publication in PLOS Digital Health.

Best regards,

Henry Horng-Shing Lu

Section Editor

PLOS Digital Health

**Additional Editor Comments (if provided):**

**Reviewer Comments (if any, and for reference):**

Reviewer's Responses to Questions

**Comments to the Author**

Reviewer #2: All comments have been addressed

Reviewer #3: All comments have been addressed

Reviewer #5: All comments have been addressed

publication criteria?

Reviewer #2: Yes

Reviewer #3: Yes

Reviewer #5: Yes

3. Has the statistical analysis been performed appropriately and rigorously?

Reviewer #2: Yes

Reviewer #3: Yes

Reviewer #5: Yes

4. Have the authors made all data underlying the findings in their manuscript fully available (please refer to the Data Availability Statement at the start of the manuscript PDF file)?

Reviewer #2: Yes

Reviewer #3: Yes

Reviewer #5: Yes

5. Is the manuscript presented in an intelligible fashion and written in standard English?

Reviewer #2: Yes

Reviewer #3: Yes

Reviewer #5: Yes

Reviewer #2: Thank you for your careful and thorough revision of the manuscript! The responses to my suggestions were detailed, and the revised manuscript has been strengthened substantially.

I appreciate the additions and clarifications around (i) time-dependent calibration (ii) reproducibility (iii) clearer justification for using XGboost and lastly (iv) more careful and explicit framing of treatment-related findings as prognostic associations rather than causal effects. The added elements such as temporal validation, expanded discussion of clinical utility, as well as inclusion of clinical vignettes further improve interpretability and translational relevance.

Overall, the manuscript now reads as methodologically rigorous, clinically meaningful, and appropriately cautious in interpretation. I have no further substantive concerns. I recommend acceptance.

Reviewer #3: Recommendation: Accept

Comments to the Authors: I have reviewed the revised manuscript and the authors responses. I appreciate the authors' diligence in addressing the concerns raised in the previous round of review. The manuscript has been substantially improved.

Specifically:

Generalizability & Validation: The addition of temporal validation using the 2022-2023 cohort (achieving stable AUCs of 0.858 and 0.878) significantly strengthens the study's validity. The authors have also appropriately acknowledged the need for future geographic/cross-ethnic validation in the Discussion.

Deep Learning vs. Tree-based Models: I agree with the revised discussion regarding the performance comparison. Acknowledging that the "simpler neural network architectures" might be a factor in the lower performance compared to XGBoost is a fair and balanced assessment.

Clinical Interpretability: The added biological and clinical context for the non-linear relationships (e.g., the U-shaped risk for BMI, the survivor bias in long-term smokers, and the age-dependent mortality acceleration) makes the SHAP analysis much more meaningful for clinical readers.

Censoring and Data Transparency: The addition of Supplementary Table S2 successfully clarifies the proportion of censored observations (alive patients ~16.5-21.2%).

Note: I reviewed Table S2 and observed a high missing rate for Tumor Grade (>52%). While this is understandable given that only ~33-35% of patients underwent surgery (and thus had histological grading), and the authors used MICE for imputation, the transparency provided in this table is crucial for readers to understand the dataset's characteristics.

Conclusion: The authors have satisfactorily addressed all my major and minor comments. The inclusion of temporal validation and the detailed supplementary data regarding censoring and missing values have addressed the robustness and transparency concerns. I have no further requests.

Reviewer #5: All concerns have been well-addressed. I recommend this manuscript to be accepted.

**Do you want your identity to be public for this peer review?** For information about this choice, including consent withdrawal, please see our Privacy Policy

Reviewer #2: **Yes:** Tuan Bao Vinh

Reviewer #3: No

Reviewer #5: No
